# Primary Stereotactic Radiosurgery Provides Favorable Tumor Control for Intraventricular Meningioma: A Retrospective Analysis

**DOI:** 10.3390/jcm12031068

**Published:** 2023-01-30

**Authors:** Motoyuki Umekawa, Yuki Shinya, Hirotaka Hasegawa, Atsuto Katano, Nobuhito Saito

**Affiliations:** 1Department of Neurosurgery, The University of Tokyo Hospital, Tokyo 113-8655, Japan; 2Department of Radiology, The University of Tokyo Hospital, Tokyo 113-8655, Japan

**Keywords:** intraventricular meningioma, stereotactic radiosurgery, long-term outcomes

## Abstract

The surgical resection of intraventricular meningiomas (IVMs) remains challenging because of their anatomically deep locations and proximity to vital structures, resulting in non-negligible morbidity and mortality rates. Stereotactic radiosurgery (SRS) is a safe and effective treatment option, providing durable tumor control for benign brain tumors, but its outcomes for IVMs have rarely been reported. Therefore, the goal of the present study was to evaluate the SRS outcomes for IVMs at our institution. This retrospective observational study included 11 patients with 12 IVMs with a median follow-up period of 52 months (range, 3–353 months) treated with SRS using the Leksell Gamma Knife. Nine (75%) tumors were located in the trigone of the lateral ventricle, two (17%) in the body of the lateral ventricle, and one (8%) in the third ventricle. Tumor control was achieved in all cases, and seven (55%) decreased in size. Post-SRS perifocal edema was observed in four (37%; three asymptomatic and one symptomatic but transient) patients, all of which were resolved by the last follow-up. SRS appears to provide safe and excellent tumor control for IVMs. A longer follow-up with a larger number of cases is desired for a more solid conclusion.

## 1. Introduction

Meningiomas are the most common benign intracranial tumors originating from the arachnoid cap cells [1,2] irrespective of the location. The standard therapeutic option is surgical resection; superficial tumors are easy to resect, while deep tumors are often challenging because of the important anatomical structures surrounding them.

Intraventricular meningiomas (IVMs) are rare, accounting for only 0.3–5% of all meningiomas. IVMs are one of the most challenging tumors because of their deep locations [3,4]. Smaller IVMs are usually asymptomatic, whereas larger IVMs can manifest various symptoms such as headache, visual field deficits, ataxia, paresis, seizure, and hydrocephalus [3,5,6,7,8,9,10]. Many factors can complicate safe surgical resection and jeopardize patients’ neurological outcomes: (1) sacrificing the cerebral cortex to approach the tumor, (2) critical nerve tracts surrounding the tumor and surgical trajectory, (3) difficulty with hemostasis deep inside the brain, especially for large tumor cases. Recent advances in neuroendoscopic surgery have offered adequate surgical exposure with minimal invasiveness, though the surgical complication rate is reportedly high—up to 33% [11,12,13,14]. Moreover, there is a non-negligible risk of surgery-associated mortality, which was reported to be 1.6% in a recent systematic review [4].

In light of the above, tumor control that preserves the surrounding functional anatomies is crucial and desirable in the management of IVMs. Stereotactic radiosurgery (SRS) is characterized by its accurate targeting and delivery of high-dose focused irradiation in a single session, offering a minimally invasive treatment option for intracranial tumors. Since radiation exposure to the surrounding structures can be adequately reduced owing to its sharp dose fall-off, SRS can be considered an appropriate treatment for IVMs; however, there remains a paucity of evidence, likely because of the rarity of IVMs. Hence, we conducted the present study including detailed analyses on the radiosurgical outcomes of IVMs to elucidate the efficacy and safety of SRS for IVMs.

## 2. Materials and Methods

### 2.1. Patient Data Collection

Of 352 patients with intracranial meningioma who underwent SRS from 1990 to 2022 at our institution, data on 12 patients with 13 IVMs were collected from the institutional gamma knife database. One patient with <3 months of follow-up was excluded from the analysis, while patients with neurofibromatosis type 2 (NF2) were included. All tumors were diagnosed based on their radiologic findings, and all radiologic images were reviewed by two independent neuroradiologists and attending neurosurgeons. The study was approved by the Institutional Review Board of our institution (#2231). All patients provided written informed consent for study participation.

### 2.2. SRS Procedure

The Leksell Gamma Knife (Elekta Instruments, Stockholm, Sweden) was used for all SRS procedures. The detailed treatment process has been previously reported [15,16]. After head fixation using a Leksell frame (Elekta Instruments), stereotactic imaging (computed tomography [CT] before July 1996, magnetic resonance imaging [MRI] between August 1996 and January 2018, followed by cone-beam CT) was performed to obtain precise tumor data. Dedicated neurosurgeons and radiation oncologists performed radiosurgical planning using commercially available software (KULA planning system until 1998 and Leksell Gamma Plan thereafter [Elekta Instruments]). In principle, 16 Gy before 2010 and 14 Gy thereafter were administered to the tumor margin using a 50 ± 5% isodose line. Representative cases are shown in Figure 1.

### 2.3. Follow-Up and Clinical Outcomes

After SRS, MRIs were checked every 6 months for the first 3 years and annually thereafter. Tumor response after SRS was judged by the Response Assessment in Neuro-Oncology criteria [17,18]; tumor progression was defined as an enlargement in volume of ≥25% upon two or more consecutive post-SRS images. Patients’ neurological status and radiological responses to SRS were prospectively collected at each hospital visit, and any adverse events were recorded based on a Common Terminology Criteria for Adverse Events (CTCAE, version 5.0) grade. Data on patients who dropped out of regular follow-ups or returned to referring physicians were collected via telephone, and follow-up radiographic images were obtained.

### 2.4. Statistical Analysis

First, the baseline characteristics of the patients were summarized. Second, progression-free rates (PFRs), disease-specific survival (DSS), overall survival (OS), neurological preservation, and post-SRS peritumoral T2 signal change rates were calculated using the Kaplan–Meier method, excluding the patient with only a follow-up of three months. Third, factors associated with PFRs and post-SRS peritumoral T2 signal change rates were examined using bivariate Cox proportional hazard analyses. Continuous variables were entered into models after being dichotomized using their median values. Statistical analyses were performed using JMP Pro 16 software (SAS Institute Inc., Cary, NC, USA).

## 3. Results

### 3.1. Patient and Tumor Characteristics

Eleven (6 women and 5 men) patients with a median age of 45 years (range, 13–80 years) were included in the study. The median post-SRS follow-up period was 52 months. The baseline characteristics and treatment data are summarized in Table 1, and the details of the patients are described in Table 2. Five patients had multiple NF2-related intracranial meningiomas and underwent SRS for IVMs. There was one patient in whom bilateral trigonal meningiomas were simultaneously treated with SRS (Table 2, No. 7 and 8). No patients had undergone prior surgery for IVMs.

### 3.2. Tumor Control

Of the 12 tumors, seven (58%) decreased in size by the last follow-up visit, while five (42%) were stable in size. Tumor control was achieved in all patients; therefore, the cumulative 5- and 10-year PFR were 100% (Figure 2). No significant differences in tumor control were observed between the sporadic and NF2-related IVMs. After SRS, five patients with multiple meningiomas underwent additional interventions for growing meningiomas other than IVMs. Eventually, two of the five NF2 patients died of progression of such tumors, although IVMs were well controlled after SRS. As a result, the cumulative 5- and 10-year DSS rates of IVMs after SRS were 100%, although the 3- and 10-year OS rates were 86% and 71%, respectively (Figure 3A). OS rates in NF2 patients were lower in NF2 patients compared with sporadic IVM patients (67% vs. 100% at 3 years, and 33% vs. 100% at 10 years, respectively), although these differences were not statistically significant (Log-rank test, *p* = 0.070; Figure 3B).

### 3.3. Adverse Radiation Events (AREs)

No AREs were observed, and the 5- and 10-year neurological preservation rates were 100% (Figure 4A). Post-SRS peritumoral T2 signal change was observed in four (33%) patients with trigonal IVM. The signal change developed at 6–29 months after SRS and diminished at 9–40 months. The 1- and 3-year cumulative post-SRS signal change rates were 18% and 40%, respectively (Figure 4B). One patient (8%) complained of a transient headache along with signal change, but her symptom and the signal change disappeared following oral administration of corticosteroid for 1 month. No factors were significantly associated with post-SRS signal change (Table 3). No other AREs, including hydrocephalus, seizure, and visual field deficit, were observed after SRS.

## 4. Discussion

In this study, we analyzed the radiosurgical outcomes of SRS for IVMs. SRS provided an excellent PFR (100% with a median follow-up period of 52 months). Importantly, our patients included NF2-related IVMs, suggesting that SRS is effective regardless of *NF2* mutation status. Transient post-SRS signal change occurred in 33% of the cases, but there were no permanent AREs. The results were promising and comparable to the SRS outcomes for intracranial meningiomas of other locations [19,20,21,22,23].

For IVM, surgical resection is more complicated than for meningiomas of other locations due to its deep location with limited accessibility and adjacent eloquent neurovascular structures, leading to high post-surgical morbidity and mortality rates. Trigonal IVMs are especially challenging among lateral ventricle IVMs because the medial part of the tumor is in contact with the optic radiation, and the feeding arteries arise from the deepest part of the tumor via the transparietal transcortical route; therefore, the reported surgical morbidity rates range from 12.5 to 60%, including hemianopia, hemiparesis, intracranial hemorrhage, and intracranial hypertension [3,11,12,24,25]. Furthermore, the surgical morbidities are reported to be more frequent and more severe in third ventricle IVMs than in lateral ventricle IVMs due to the proximity to the thalamus, brainstem, and cranial nerve nuclei [25,26,27,28,29,30]. As a result, the mortality rate is reportedly high up to 4%, 44% of which occurred during the postoperative period [4].

On the other hand, there are only a few previous studies describing outcomes of SRS for IVMSs, with relatively favorable tumor control rates ranging from 67% to 100% reported. (Table 4; We searched PubMed without language restrictions for papers published from database inception up to December 1st, 2022 to include studies of intraventricular meningioma. We used the search terms “intraventricular, meningioma” “ventricular meningioma” “stereotactic radiosurgery”. We identified 127 previous reports about IVMs including five studies about validating SRS for IVMs.) [19,20,21,22,23]. Although our study demonstrated excellent tumor control, some previous studies revealed that salvage SRS for progressive recurrent tumors may not always provide sufficient tumor control. In the studies conducted by Kim et al. [20] and Daza-Ovalle et al. [19], two of the three (67%) failed cases were salvage cases for recurrent tumors following prior resection. This suggests that a radiation dose may need to be increased for such progressive tumors. From another viewpoint, upfront SRS for residual tumors may be more reasonable than salvage SRS after recurrence, as was proved in a recent multi-center retrospective study [31]. Notably, all the tumors in patients who had NF2 or multiple meningiomas were under good control after SRS. Although patients with multiple meningiomas tend to require multiple interventions and exhibit a shorter overall PFR [32,33,34], NF2-associated IVMs could be controlled with SRS [23]. Further case accumulation is desirable for a more solid conclusion on these issues.

Newly or worsening perifocal edema is the main ARE in SRS for IVMs [19,20,21,22,23]. In general, peritumoral T2 signal changes occur in 28–50% of cases after SRS for meningiomas, with the incidence of symptomatic ones ranging between 5% and 43% [35]. The risks of symptomatic signal change include older age, larger tumor volume, higher radiosurgical dose, presence of peritumoral edema before SRS, and primary SRS [35,36]. The present study observed post-SRS signal changes in four tumors; three (25%) were asymptomatic and one (8%) was symptomatic but transient. Previous reports on SRS for IVMs have shown that an incidence of post-SRS peritumoral edema ranges from 17% and 100% [19,20,21,22,23]. In a report by Nundkumar et al. in which two of their two cases developed post-SRS peritumoral edema, a marginal dose as high as 18 Gy was used, and surgical resection was required due to the uncontrolled edema even without tumor progression in one of them [22]. Given, as Daza-Ovalle et al. highlight, the association between volume, received >12 Gy irradiation, and the occurrence of peritumoral edema [19], radiosurgical dose may be an important determiner and needs to be balanced according to tumor volume. Despite a certain risk of perifocal edema, all of our patients went through it without surgical intervention, developing no permanent morbidity.

The optimal radiosurgical dose for IVMs remains debatable. As shown in Table 4, 12–18 Gy was mainly used as a margin dose. Nevertheless, 18 Gy appears to be too high for a margin dose, given that one of the two patients in our study and two of the two patients in Nundkumar’s cohort [22] who received 18 Gy at the tumor margins later developed perifocal edema. Initially, 16 Gy was used as a margin dose in our institution, but this has currently been reduced to 14 Gy in order to reduce RAE. At this moment, the optimal radiosurgical dose would be somewhere between 12 and 16 Gy. As in the previous discussion, a higher dose may be desirable for progressive recurrent tumors.

This study has several limitations. First, it was a retrospective, single-institutional study with a potential selection bias. To determine the efficacy and safety of SRS for larger tumors, further investigation is required. Second, all tumors in this study were radiologically diagnosed; therefore, the diagnoses might have been less reliable than a histological diagnosis. A larger number of patients would be desirable for future studies to re-confirm our results.

## 5. Conclusions

SRS can be an appropriate treatment option for IVMs, achieving favorable mid-term tumor control without jeopardizing neurological function. Further investigation in a larger volume study is warranted to establish the role of SRS for IVMs.

## Figures and Tables

**Figure 1 jcm-12-01068-f001:**
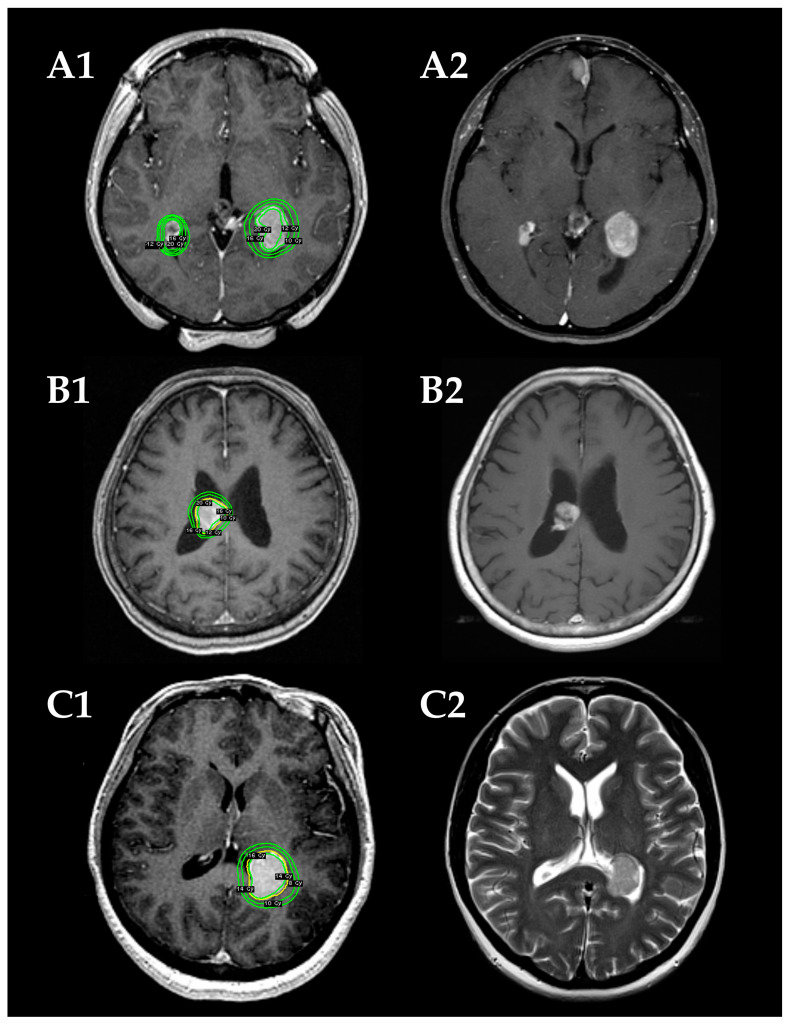
Radiosurgical plans (**A1**–**C1**) and follow-up magnetic resonance images (MRIs) (**A2**–**C2**) in three demonstrative patients. (**A**) A 31-year-old woman with NF2-related bilateral intraventricular meningiomas (IVMs) in the trigones of the lateral ventricles (cases 7 and 8). The tumors are stable in size on a follow-up MRI at 21 months after the stereotactic radiosurgery (SRS). (**B**) An 80-year-old woman with a sporadic IVM in the body of the right lateral ventricle (case 9). Tumor shrinkage is confirmed on a follow-up MRI at 109 months after the SRS. (**C**) A 50-year-old woman with a sporadic IVM in left trigone (case 11). Tumor shrinkage is confirmed on a follow-up MRI at 35 months after the SRS.

**Figure 2 jcm-12-01068-f002:**
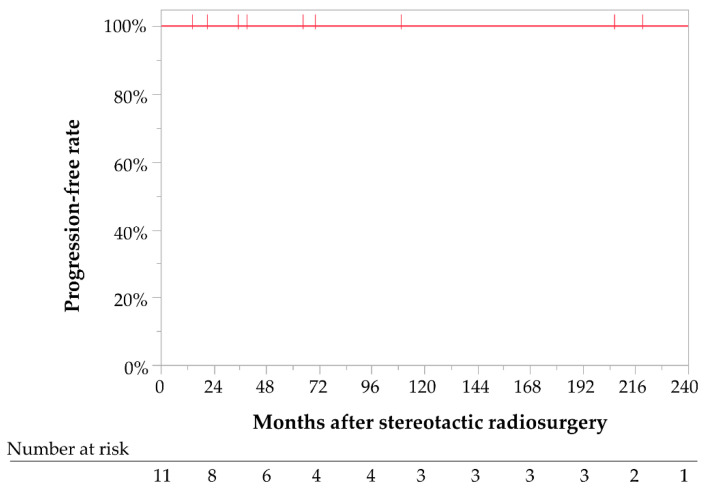
Kaplan–Meier curve for progression-free rates for all patients with intraventricular meningiomas treated with stereotactic radiosurgery.

**Figure 3 jcm-12-01068-f003:**
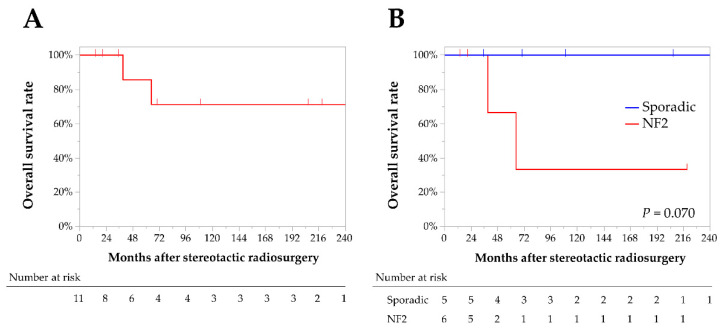
Kaplan–Meier curves for the overall survival rates in all patients (**A**), and a comparison between sporadic and neurofibromatosis-type-2-related intraventricular meningiomas (**B**).

**Figure 4 jcm-12-01068-f004:**
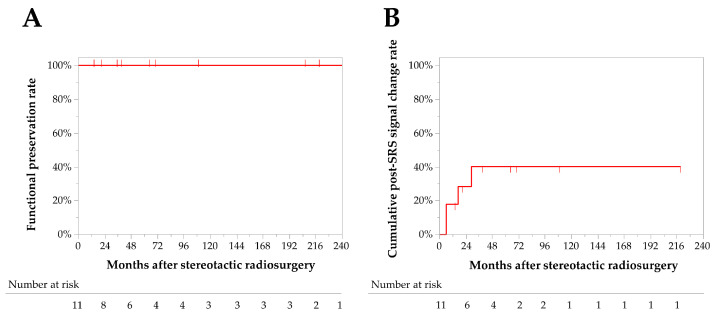
Kaplan–Meier curves for functional preservation rates (**A**) and cumulative post-stereotacitic radiosurgery T2 signal change rates (**B**).

**Table 1 jcm-12-01068-t001:** Patients’ baseline characteristics and treatment data of stereotactic radiosurgery.

Variables	
**Median (Range)**	**Tumors (*n* = 12)**
Follow-up period, months	52 (3–353)
Age at SRS, years	45 (13–80)
Maximum diameter, mm	24 (17–33)
Target volume, mL	4.9 (1.2–9.8)
Marginal dose, Gy	16 (9–18)
Central dose, Gy	34 (24–45)
Number of isocenters	10 (3–37)
Volume of normal brain tissue exposed to ≥12 Gy (V12), mL	1.7 (1.2–2.5)
***n* (%)**	**Patients (*n* = 11)**
Male sex	5 (45)
Prior surgery for IVMs	0 (0)
Multiple meningiomas	6 (55)
NF2	5 (45)
***n* (%)**	**Tumors (*n* = 12)**
Brain edema before SRS	3 (25)
Tumor location	
- Trigone	9 (75)
- Body of lateral ventricle	2 (17)
- Third ventricle	1 (8)

*n*, number; IVM, intraventricular meningioma; NF2, neurofibromatosis type 2; SRS, stereotactic radiosurgery.

**Table 2 jcm-12-01068-t002:** Summary data of all intraventricular meningiomas treated with stereotactic radiosurgery.

No.	Age/Sex	Location	Tumor Features	Prior Surgery	Tumor Volume	Marginal Dose	Central Dose	Follow-Up	Tumor Size at the Last F/U	ARE	Patients’ Status at the Last F/U
1	22/F	Trigone	Sporadic	None	7.4 mL	9 Gy	45 Gy	353 months	Shrinkage	Signal change (asymptomatic)	Alive
2	52/F	Trigone	Sporadic	None	1.8 mL	18 Gy	45 Gy	206 months	Shrinkage	Signal change (asymptomatic)	Alive
3	13/M	Body of the LV	NF2	None	2.9 mL	16 Gy	40 Gy	14 months	Shrinkage	None	Alive
4	46/F	Trigone	Sporadic	None	7.5 mL	18 Gy	36 Gy	70 months	Shrinkage	None	Alive
5	44/M	Third ventricle	NF2	None	2.5 mL	15.5 Gy	24 Gy	219 months	Shrinkage	None	Alive
6	33/M	Trigone	NF2	None	9.8 mL	14 Gy	28 Gy	65 months	Unchanged	None	Died of progression of an unrelated tumor
7	31/F	Trigone	NF2	None	5.5 mL	16 Gy	32 Gy	21 months	Unchanged	Signal change (headache)	Alive
8	31/F	Trigone	NF2	None	1.2 mL	16 Gy	32 Gy	21 months	Unchanged	None	Alive
9	80/F	Body of the LV	Sporadic	None	4.3 mL	16 Gy	40 Gy	109 months	Shrinkage	None	Alive
10	67/M	Trigone	NF2	None	9.0 mL	14 Gy	35 Gy	39 months	Unchanged	None	Died of progression of an unrelated tumor
11	50/F	Trigone	Sporadic	None	8.3 mL	14 Gy	28 Gy	35 months	Shrinkage	Signal change (asymptomatic)	Alive
12	63/M	Trigone	Sporadic	None	4.0 mL	14 Gy	28 Gy	3 months	Unchanged	None	Alive

ARE, adverse radiation effect; F, female; IVM, intraventricular meningioma; LV, lateral ventricle; NF2, neurofibromatosis type 2; M, male.

**Table 3 jcm-12-01068-t003:** Bivariate Cox proportionate analysis on post-SRS signal change.

Variables	HR (95% CI)	*p*-Value
Patient age (continuous)	0.98 (0.92–1.03)	0.478
Patient age ≥ 45 (vs. <45)	1.10 (0.15–7.79)	0.927
Maximum diameter, mm (continuous)	0.99 (0.79–1.22)	0.961
Maximum diameter ≥ 25 mm (vs. <25)	2.34 (0.24–22.80)	0.464
Tumor volume, mL (continuous)	1.09 (0.79–1.64)	0.600
Tumor volume ≥ 8 mL (vs. <8)	1.48 (0.21–10.63)	0.697
Marginal dose, Gy (continuous)	0.91 (0.66–1.36)	0.605
Marginal dose ≥ 16 Gy (vs. <16 Gy)	1.08 (0.15–7.92)	0.939
Central dose, Gy (continuous)	0.05 (0.91–1.25)	0.470
Central dose ≥ 33 Gy (vs. <33 Gy)	0.68 (0.09–4.85)	0.697
Number of isocenters (continuous)	1.01 (0.76–1.33)	0.962
Number of isocenters ≥ 7 (vs. <7)	0.77 (0.11–5.55)	0.794
V12, mL (continuous)	1.44 (0.02–71.90)	0.847
V12 ≥ 2.0 mL (vs. <2.0 mL)	1.67 (0.10–26.65)	0.718
Brain edema before SRS (vs. without edema)	2.56 (0.36–18.25)	0.349
NF2-related (vs. sporadic)	0.30 (0.03–2.92)	0.302

CI, confidence interval; HR, hazard ratio; NF2, neurofibromatosis type 2; SRS, stereotactic radiosurgery; V12, volume of normal brain tissue exposed to ≥12 Gy.

**Table 4 jcm-12-01068-t004:** Summary of previous studies of SRS for IVMs.

Authors	Number of Tumors	Median Age, Years (Range)	Male (%)	NF2 (%)	Location (%)	Prior Resection (%)	Median Tumor Volume, mL (Range)	Median Margin Dose, Gy (Range)	Median Follow-Up Months (Range)	Tumor Control Rate (%)	Post-SRS Signal Change (Symptomatic) (%)	Post-SRS Hydrocephalus (%)
Kim et al., 2009 [27]	9	51 (14–81)	67	0	Trigone 89 Third ventricle 11	44	3.9 (0.8–11.8)	16 (14–25)	64 (7–161)	67	0	0
Nundkumar et al., 2013 [29]	2	50 (49–50)	0	NA	Trigone 100	0	3.3 (2.2–4.4)	18 (18–18)	12 (8–17)	100	100 (100)	0
Mindermann et al., 2020 [28]	5	63 (50–81)	0	NA	Trigone 100	0	4.7 (2.5–14.1)	13.5 (12–15)	81 (19–240)	100	80 (40)	0
Samanci et al., 2020 [30]	6	41 (30–71)	50	17	Trigone 100	17	5.5 (1.2–9.2)	12 (11–13)	74 (24–139)	100	17 (17)	0
Daza-Ovalle et al., 2022 [26]	20	53 (14–84)	63	0	Trigone 90 Third ventricle 5 Fourth ventricle 5	30	4.8 (0.8–17)	14 (12–25)	63 (6–322)	85	35 (15)	0
**Present study 2022**	**12**	**45** **(13–80)**	**45**	**50**	**Trigone 75** **Body of lateral ventricle 17** **Third ventricle 8**	**0**	**4.9** **(1.2–9.8)**	**16** **(9–18)**	**52** **(3–353)**	**100**	**33** **(8)**	**0**

## Data Availability

Anonymized data in this article will be available by request from any qualified investigator, and information about the method of analysis will be available from the corresponding author upon reasonable request.

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
