# Peer review of "Primary Stereotactic Radiosurgery Provides Favorable Tumor Control for Intraventricular Meningioma: A Retrospective Analysis"

_jcm, 2023, doi:10.3390/jcm12031068_

Round 1

Reviewer 1 Report

SRS in the treatment of intracranial meningiomas is a subject extensively covered in the literature. Indeed, intraventricular meningiomsa are rarely described. The results of the study are clearly presented. It is not clear whether the diagnosis was obtained radiologically or by biopsy. A comparison with populations subjected to surgical resection would also have been interesting, in order to evaluate their morbidity. As rightly written by the authors, larger numbers are needed (but the disease itself is rare) and longer follow-up periods. Furthermore, the small size of the population does not allow for cohort analyses.

Author Response

Dear Reviewer #1,

We really appreciate your valuable comments about our manuscript. Please find below your itemized questions and concerns, followed by our replies in blue. These responses are also uploaded as an attachment.

Sincerely yours,

Motoyuki Umekawa

SRS in the treatment of intracranial meningiomas is a subject extensively covered in the literature. Indeed, intraventricular meningiomas are rarely described. The results of the study are clearly presented. It is not clear whether the diagnosis was obtained radiologically or by biopsy.

Thank you for your valuable comments, and we agree with your points. In light of the diagnosis, all tumors in this study were diagnosed based on their radiological findings. In NF2 patients, tumors were diagnosed based on other meningiomas’ pathological findings. Based on the above facts and your suggestion, we have corrected the description below to illustrate the tumor diagnosis more clearly.

[Page 2, lines 57–59]

All tumors were diagnosed based on their radiologic findings, and all radiologic images were reviewed by two independent neuroradiologists and attending neurosurgeons.

A comparison with populations subjected to surgical resection would also have been interesting, in order to evaluate their morbidity.

We appreciate your comment. Unfortunately, we could not compare this SRS case series with the surgery cohort because we did not have any IVM patients treated with surgery. As an alternative, we have reviewed previous studies on surgery for IVMs and discussed the pros and cons of surgery vs. SRS for IVM as follows.

[Page 8, lines 172–183]

For IVM, surgical resection is more complicated than for other meningiomas in other locations due to its deep location with limited accessibility and adjacent eloquent neurovascular structures, leading to high post-surgical morbidity and mortality rates. Trigonal IVMs are especially challenging among lateral ventricle IVMs because the medial part of the tumor is in contact with the optic radiation, and the feeding arteries arise from the deepest part of the tumor via the transparietal transcortical route; therefore, the reported surgical morbidity rates range from 12.5 to 60%, including hemianopia, hemiparesis, intracranial hemorrhage, and intracranial hypertension [3, 11, 12, 19, 20]. Furthermore, the surgical morbidities are reported to be more frequent and more severe in third ventricle IVMs than in lateral ventricle IVMs due to the proximity to the thalamus, brainstem, and cranial nerve nuclei [20-25]. As a result, the mortality rate is reportedly high up to 4%, 44% of which occurred during the postoperative period [4].

As rightly written by the authors, larger numbers are needed (but the disease itself is rare) and longer follow-up periods. Furthermore, the small size of the population does not allow for cohort analyses.

Thank you for your crucial comment on this study design. As you pointed out, the number of patients (11 patients with 12 tumors) in this study was relatively small because of the rarity of IVMs; therefore, may not be appropriate for a cohort study. Based on your comment, we have re-analyzed the data and revised our manuscript as follows.

[Page 6, lines 141–142]

Figure 2. Kaplan–Meier curve for progression-free rates for the entire patients of intraventricular meningiomas treated with stereotactic radiosurgery.

[Page 7, lines 144–146]

Figure 3. Kaplan–Meier curves for the overall survival rates in the entire patients (A), and a comparison between sporadic and neurofibromatosis type 2-related intraventricular meningiomas (B).

[Page 8, lines 167–169]

Importantly, our patients included NF2-related IVMs, suggesting that SRS is effective regardless of NF2 mutation status.

[Page 11, lines 225–228]

Nevertheless, 18 Gy appears to be too high for margin dose, given that one of the two patients in our study and two of the two patients in Nundkumar’s cohort [29] who received 18 Gy at the tumor margins later developed perifocal edema.

[Page 11, lines 235–237]

Second, all tumors in this study were radiologically diagnosed; therefore, the diagnoses might have been less reliable than histological diagnosis.

Reviewer 2 Report

The authors present a retrospective study investigating primary stereotactic radiosurgery for intraventricular meningioma. Intraventricular meningiomas are a rare anatomical subtype of cranial meningiomas and surgical resection can be more challenging compared to the majority of cranial meningiomas. Hence, the authors address an essential issue. In the following I summarize some issues regarding the present manuscript:

-       Introduction: The authors provide adequate references, and this section introduces the readership to the present topic.

-       Methods: The authors should provide a reference for their definition of meningioma progression

-       Results: I appreciate the interesting clinical approach. However, there is a wide range of follow-up. One patient had only a follow-up of three months in this series of 11 patients. Hence, I think it is not appropriate to draw a conclusion from the Kaplan-meier curves. Furthermore, the authors should also evaluate the risk factors (tumor volume, brain edema before radiosurgery, marginal dose, number of isocenters, volume of normal brain tissue exposed to ≥12Gy) regarding the development of a peritumoral edema after stereotactic radiosurgery for intraventricular meningiomas. 

-       Discussion: The authors provide a summary of previous studies investigating stereotactic radiosurgery for intraventricular meningiomas. The authors should also clarify the literature search workflow (e.g., databases, keywords, inclusion criteria).

Author Response

Dear Reviewer #2,

We really appreciate your encouraging comment. We have revised the manuscript given your crucial comments. Please find below the itemized questions and concerns mentioned in the review, followed by our replies in blue.

These responses are also uploaded as an attachment.

Sincerely yours,

Motoyuki Umekawa

-       Introduction: The authors provide adequate references, and this section introduces the readership to the present topic.

Thank you for your thoughtful comment.

-       Methods: The authors should provide a reference for their definition of meningioma progression

Thank you for your suggestion. Based on your comment, we have added references as follows.

[Page 3, lines 88–90]

Tumor response after SRS was judged by the Response Assessment in Neuro-Oncology criteria [17, 18]; tumor progression was defined as an enlargement in volume of ≥ 25% upon two or more consecutive post-SRS images.

  1. Wen, P. Y., D. R. Macdonald, D. A. Reardon, T. F. Cloughesy, A. G. Sorensen, E. Galanis, J. Degroot, W. Wick, M. R. Gilbert, A. B. Lassman, et al. "Updated response assessment criteria for high-grade gliomas: Response assessment in neuro-oncology working group." J Clin Oncol 28 (2010): 1963-72. 10.1200/JCO.2009.26.3541. https://www.ncbi.nlm.nih.gov/pubmed/20231676.
  2. Huang, R. Y., W. L. Bi, M. Weller, T. Kaley, J. Blakeley, I. Dunn, E. Galanis, M. Preusser, M. McDermott, L. Rogers, et al. "Proposed response assessment and endpoints for meningioma clinical trials: Report from the response assessment in neuro-oncology working group." Neuro Oncol 21 (2019): 26-36. 10.1093/neuonc/noy137. https://www.ncbi.nlm.nih.gov/pubmed/30137421.

-       Results: I appreciate the interesting clinical approach. However, there is a wide range of follow-up. One patient had only a follow-up of three months in this series of 11 patients. Hence, I think it is not appropriate to draw a conclusion from the Kaplan-meier curves.

We appreciate your valuable comments. According to your advice, we re-analyzed Kaplan-Meier method and re-calculated p-value excluding the patient with only a follow-up of three months. The sentence in the Materials and Method section and Figures have been revised as follows.

[Page 3, lines 98–101]

Second, progression-free rates (PFRs), disease-specific survival (DSS), overall survival (OS), neurological preservation, and post-SRS peritumoral T2 signal change rates were calculated using the Kaplan–Meier method, excluding the patient with only a follow-up of three months.

Figure 2

Figure 3

Figure 4

Furthermore, the authors should also evaluate the risk factors (tumor volume, brain edema before radiosurgery, marginal dose, number of isocenters, volume of normal brain tissue exposed to ≥12Gy) regarding the development of a peritumoral edema after stereotactic radiosurgery for intraventricular meningiomas. 

Thank you for your valuable comments.

Based on your suggestion, we have collected the patients’ data on brain edema before SRS, the number of isocenters, and 12Gy volume. And re-analyzed the associated risk factors with post-SRS signal change, including these factors. We have added these results in our manuscript as follows.

[Table 1]

Table 1. Patients’ baseline characteristics and treatment data of stereotactic radiosurgery.

[Page 7, lines 155–156]

No factors were significantly associated with post-SRS signal change (Table 3).

[Table 3]

Table 3. Bivariate Cox proportionate analysis on post-SRS signal change.

-       Discussion: The authors provide a summary of previous studies investigating stereotactic radiosurgery for intraventricular meningiomas. The authors should also clarify the literature search workflow (e.g., databases, keywords, inclusion criteria).

We appreciate your thoughtful comment. Given your proposal, we have added the description in the discussion section as follows.

[Page 8, lines 184–190]

On the other hand, there are only a few previous studies describing outcomes of SRS for IVMSs, with relatively favorable tumor control rates ranging from 67% to 100% reported (Table 4; We searched PubMed without language restrictions for papers published from database inception up to December 1st, 2022, to include studies of intraventricular meningioma. We used the search terms "intraventricular, meningioma" "ventricular meningioma," "stereotactic radiosurgery". We identified 127 previous reports about IVMs including 5 studies about validating SRS for IVMs.) [26-30].

Round 2

Reviewer 1 Report

I would like to thank the authors for their efforts to improve the quality and structure of the paper, which is now more outlined and precise in the endo-points. The difference encountered between sporadic IVM and NF is interesting, which reinforces what we already know about neurofibromatosis. The number of cases and the follow-up time are unfortunatly small, but it is clearly stated as a bias by the authors at the end of the work. All in all, my judgment is positive and I think that this work can add something to the topic of radiosurgery of intraventricular meningiomas, rarely described in literature.

Reviewer 2 Report

The authors addressed all issues.